# Neuroanatomical Quantitative Proteomics Reveals Common Pathogenic Biological Routes between Amyotrophic Lateral Sclerosis (ALS) and Frontotemporal Dementia (FTD)

**DOI:** 10.3390/ijms20010004

**Published:** 2018-12-20

**Authors:** Marina Oaia Iridoy, Irene Zubiri, María Victoria Zelaya, Leyre Martinez, Karina Ausín, Mercedes Lachen-Montes, Enrique Santamaría, Joaquín Fernandez-Irigoyen, Ivonne Jericó

**Affiliations:** 1Department of Neurology ComplejoHospitalario de Navarra (CHN), IdiSNA (Navarra Institute for Health Research), Irunlarrea 3, 31008 Pamplona, Spain; lmmerino@hotmail.com; 2Proteored-ISCIII, Proteomics Unit, Navarrabiomed, Complejo Hospitalario de Navarra (CHN), Universidad Pública de Navarra (UPNA), IdiSNA, Irunlarrea 3, 31008 Pamplona, Spain; irene.zubiri.azcarate@navarra.es (I.Z.); karina.ausin.perez@navarra.es (K.A.); mercedes.lachen.montes@navarra.es (M.L.-M.); enrique.santamaria.martinez@navarra.es (E.S.); joaquin.fernandez.irigoyen@navarra.es (J.F.-I.); 3Pathological Anatomyservice Complejo Hospitalario de Navarra (CHN), IdiSNA (Navarra Institute for Health Research), Irunlarrea 3, 31008 Pamplona, Spain; mv.zelaya.huerta@navarra.es; 4Clinical Neuroproteomics Group, Navarrabiomed, Complejo Hospitalario de Navarra (CHN), Universidad Pública de Navarra (UPNA), IdiSNA, Irunlarrea 3, 31008 Pamplona, Spain

**Keywords:** amyotrophic lateral sclerosis (ALS), frontotemporal dementia (FTD), motor neuron, proteomics

## Abstract

(1) Background: Amyotrophic lateral sclerosis (ALS) and frontotemporal dementia (FTD) are neurodegenerative disorders with an overlap in clinical presentation and neuropathology. Common and differential mechanisms leading to protein expression changes and neurodegeneration in ALS and FTD were studied trough a deep neuroproteome mapping of the spinal cord. (2) Methods: A liquid chromatography tandem mass spectrometry (LC-MS/MS) analysis of the spinal cord from ALS-TAR DNA-binding protein 43 (*TDP-43*) subjects, ubiquitin-positive frontotemporal lobar degeneration (FTLD-U) subjects and controls without neurodegenerative disease was performed. (3) Results: 281 differentially expressed proteins were detected among ALS versus controls, while 52 proteins were dysregulated among FTLD-U versus controls. Thirty-three differential proteins were shared between both syndromes. The resulting data was subjected to network-driven proteomics analysis, revealing mitochondrial dysfunction and metabolic impairment, both for ALS and FTLD-U that could be validated through the confirmation of expression levels changes of the Prohibitin (*PHB*) complex. (4) Conclusions: ALS-TDP-43 and FTLD-U share molecular and functional alterations, although part of the proteostatic impairment is region- and disease-specific. We have confirmed the involvement of specific proteins previously associated with ALS (Galectin 2 (*LGALS3*), Transthyretin (*TTR*), Protein S100-A6 (*S100A6*), and Protein S100-A11 (*S100A11*)) and have shown the involvement of proteins not previously described in the ALS context (Methanethiol oxidase (*SELENBP1*), Peptidyl-prolyl cis-trans isomerase NIMA-interacting 1 (*PIN-1*), Calcyclin-binding protein (*CACYBP*) and Rho-associated protein kinase 2 (*ROCK2*)).

## 1. Introduction

Amyotrophic lateral sclerosis (ALS) is a neurodegenerative disease that derives from a combined degeneration of upper and lower motor neurons in the spinal cord and motor cortex and follows a fatal course with a median survival time less than five years [1]. In Europe and the United States, its prevalence is 3–5 cases per 100,000 inhabitants/year [2]. This syndrome affects individuals of both genders with a higher prevalence in men than in women (1.7/1) and it manifests at any age with a peak incidence between ages 45–65 years [3]. Although it is usually a sporadic disease, 8–10% are identified as familiar [4]. Super oxide dismutase 1 (*SOD1*) was the first ALS gene to be identified in 1993, since then more than 120 genetic variants have been associated with a risk of ALS and at least 25 of these genes have been reproducibly implicated in familiar ALS with moderated penetrance, but nowadays 80% of familial cases are not linked to known genetic causes [5].

ALS diagnosis is based on clinical examination in conjunction with electromyography and laboratory testing. These tests allow ruling out other reversible disorders that may resemble ALS [6]. Patient diagnosis is based on the El Escorial criteria [7]. The clinical hallmark of ALS is the involvement of motor neurons, and the onset and early progression of ALS are frequently insidious, so symptoms may go unrecognized and undiagnosed for up to 12 months [6]. It was observed that up to 50% also have cognitive impairment of the frontal profile and 15% of patients present frontotemporal dementia (FTD), therefore the clinical spectrum of the disease goes beyond the involvement of motor neurons [8,9]. It is widely known that a common clinical spectrum between ALS and FTD, and a genetic and pathogenic overlap between both diseases has also been described [10,11]. However, in ALS there is a great heterogeneity from a neuropathological point of view and recent studies show a pathological overlapping between ALS and others neurodegenerative diseases [12]. Ubiquitin frontotemporal lobar degeneration (FTLD-U) is the most common form of frontotemporal dementia (FTD) from a neuropathological point of view and shares with some variants of ALS the aggregation and deposition of *TDP-43* immunoreactive intracytoplasmic inclusions in neurons. This pathological hallmark defines the so-called (*TDP-43* proteinopathies or tardopathies) [13,14]. Among the FTLD-U and ALS-TDP-43 are shared relevant genetic mutations, the most frequent mutation of both diseases is the expansion of the GGGGCC hexanucleotide, in the non-coding region of the *C9ORF72* gene [15,16]. In addition to this mutation there are other well described genetic alterations shared by the two diseases such as *FUS, UBQLN2, MATR3, TARDBP, VCP, TUBA4A* and *CHCHD10* [17,18,19].

Despite all the progress made in the last decade understanding, the molecular processes underlying the earliest stages and progression of these tardopathies, the origin of these devastating diseases remains unclear and the etiopathogenesis is still unknown. There are several theories regarding the biochemical mechanisms that leads to neuronal death, such as oligodendrocytic degeneration, excitotoxicity, oxidative stress, mitochondrial dysfunction, alterations in axonal transport, neuroinflammation and aberrant conformational changes of proteins among others [20,21,22]. These mechanisms could interrelate with eachother and consequently lead to the degeneration and death of the motor neuron suggesting a multistep process [23].

Neuroproteomic leads to a better understanding of the protein-driven molecular mechanisms and functions of the central nervous system (CNS) and provides the possibility of performing large-scale studies of protein functions, interactions, dynamics and structures, complements genomic and transcriptomic studies [24]. Here proteomics has been applied to study the protein expression changes in spinal cord of ALS patients and FTLD patients to identify potential biomarkers of ALS and FTD [25,26,27]. In the present study, a deep proteomic analysis of postmortem tissue of the anterior horns of the spinal cord and no-motor frontal cortex from patients with clinical and pathological diagnosis of ALS-TDP-43 and FTLD-U compared with controls without neurodegenerative diseases, has been conducted. The resulting differences have allowed us to identify significantly dysregulated proteins and processes common to both diseases and differences that are exclusively identified in one of the two entities. The present study will contribute to a deeper understanding of the disease processes and to better understand the link and the differences encompassed in the course of these neurodegenerative diseases.

## 2. Results

### 2.1. Commonalities and Differences in the Spinal Cord: Proteostatic Imbalance in ALS and FTLD-U

A total of 2318 proteins were identified in the anterior horn of the spine, of which 1002 were quantifiable. 281 proteins were differentially expressed in ALS cases when confronted to healthy control cases. However, 52 proteins showed significant differential expression between cases of FTLD-U and healthy controls (the complete list of significantly regulated proteins is presented in Appendix A). Thirty-three proteins were found to be significantly deregulated in both diseases (Figure 1). Most of the significantly dysregulated proteins were exclusively dysregulated in ALS, the 33 proteins dysregulated in both diseases represented only the 11% of the dysregulated proteins in ALS and a more relevant proportion a 60% of the significantly dysregulated proteins in FTLD-U (Table 1).

Interestingly, among the significantly dysregulated proteins in the ALS proteome, 14 have previously been proposed as potential biomarkers or relevant proteins involved in ALS. In the proteomic study 6 of these proteins were detected as up-regulated, while the other 8 were detected as significantly down-regulated in ALS (Table 2). Our proteomic data are therefore in agreement with alterations previously characterized in the ALS field. A protein panel was selected for further validation using orthogonal techniques described later on.

### 2.2. Cross-Neuroanatomical Protein Profile between ALS and FTLD-U: Region and Disease Specificities

To validate the results obtained in proteomics study and to characterize the steady-state levels of the same proteins in the target region of FTLD-U disease, we used western-blotting technique. The expression of eight proteins of interest was evaluated in spinal cord and Non motor cortex (NMC) for each of the individuals included in the discovery cohort. Subsequent experiments were performed to: (1) verify the proteomic results, re-testing the same region (spinal cord) analyzed in the discovery experiment and (2) assess the expression of the same proteins in parallel the target region for FTLD-U (the NMC). Four proteins previously reported in the literature as regulated in ALS were selected for validation: Galectin-3 (*LGALS3*), prealbumin (*TTR*), Protein S100-A11 (*S100A11*) and Protein S100-A6 (*S100A6*). In addition, four proteins not previously linked to ALS, were selected for further validation; Methanethiol oxidase (*SELENBP1*), Peptidyl-prolyl cis-trans isomerase NIMA-interacting 1 (*PIN-1*), Calcyclin-binding protein (*CACYBP*) and Rho-associated protein kinase 2 (*ROCK 2*), these proteins were found significantly dysregulated in our proteomic study. According to the regulation patterns observed in the western blotting results, the 8 proteins were classified in three differential expression profiles:Area and disease specific regulation was observed for Galectin-3 and *SELENBP1*. These two proteins showed a strong up-regulation in spinal cord for the ALS patients, while this noticeable up-regulation could only be detected in NMC for FTLD-U patients (Figure 2A). Therefore showing specific regulation in the target area for each of the diseases. *TTR*, *S100A11*, *S100A6* and *PIN1* (Figure 2B) showed ALS specific regulation. These 4 proteins were confirmed as significantly dysregulated exclusively in ALS. *TTR* was found significantly up-regulated only for ALS when analyzing the spinal cord. *S100A11* and *S100A6* were exclusively measurable in spinal cord, showing very significantly up-regulation in ALS patients and not showing relevant changes for FTLD-U patients. *PIN1* was also detected a significantly down-regulated only in ALS in booth regions. *PIN1* was observed down-regulated in ALS and FTLD-U spinal cord in the proteomic analysis, here a discrete, but not significant decrease for FTLD-U in spinal cord could be measured.Not disease or area specific protein regulation, *CACYBP* was found significantly down-regulated in spinal cord for ALS, the opposite trend was observed in the NMC, with significant up-regulation in ALS and a more drastic increase for FTLD-U patients. *ROCK 2* down-regulation was validated in both regions with a stronger down-regulation in spinal cord for both diseases, both in spinal cord and NMC the down-regulation was moderately stronger for ALS patients (Figure 2C).

### 2.3. Proteome Modules Deregulated in ALS and FTLD-U at Spinal Cord Level

To perform a proteome mapping analysis of impaired protein profiles, we used the Ingenuity Pathway Analysis (IPA) tool (Figure 3 and Figure 4). IPA uses information from experimental and predictive origin to generate pathway-specific alterations involving the deregulated proteome characterized by a proteomic analysis.

Protein interactome maps were constructed independently for each disease phenotype using the IPA software (Figure 3 and Figure 4). Network-driven proteomics revealed mitochondrial dysfunction and metabolic impairment, both for ALS (Figure 3) and FTLD-U (Figure 4). In addition dysregulated protein interactions related to nucleic acid metabolism and to energy production were found overrepresented in ALS, and the interactome map showed an enrichment in cell death and survival related protein regulation in FTLD-U. Among the dysregulated features 21 Ingenuity canonical pathways were found significantly enriched, both in ALS and FTLD-U (Appendix A) among them mitochondrial dysfunction was the most significantly enriched pathway for both diseases.

### 2.4. Network-Driven Proteomics Reveals a Common Disruption of Focal Adhesion Kinase 1/Alpha Serine/Threonine-Protein Kinase (FAK/Akt) Axis in ALS-FTD Spectrum and a Specific Non-Motor Cortical Activation of Mitogen-Activated Protein Kinase (MAPK) Route in FTLD-U

An additional aspect of interaction networks is the ability to show and highlight potentially relevant players that have gone undetected in the proteomic study. In this sense, the interaction networks reveled different cell signaling mediators including; alpha serine/threonine-protein kinase (*AKT*), Mitogen-activated protein kinase 1 (*ERK*) or Dual specificity mitogen-activated protein kinase 1 (*MAP2K*) (Figure 3 and Figure 4). The involvement of these potential candidates was considered an interesting subject for further evaluation. *AKT*, Dual specificity mitogen-activated protein kinase kinase 2 (*MEK*) and *MAP2K* were not detected in the proteomic study, while *ERK* was quantified and found up-regulated in ALS.

Subsequent experiments were performed to monitor the activation state of this kinase panel across ALS-FTLD-U spectrum in both selected brain areas spinal cord and NMC (Figure 5). *AKT* and *pAKT* showed significant down-regulation in the spinal cord for both ALS and FTDL-U cases. This event was very significant in ALS. When analyzing NMC only FTLD-U patients showed significant down-regulation of *AKT. FAK* and *p-FAK* were found significantly dysregulated and very significantly dysregulated in both diseases when measured in spinal cord, while significant down-regulation was only measured in FTLD-U when analyzing NMC.

*ERK* up-regulation was measured in the proteomic experiment, here the same trend could be appreciated by western blot, but not with statistical significance. Nevertheless p-*ERK* could be detected as significantly up-regulated for FTLD-U in NMC. *MEK* and *pMEK* significant dysregulation was detectable in NMC, in ALS for *MEK* and only in FTLD-U for *pMEK*. Suggesting differential regional implications for each disease for the different factors regulating cell signaling events (Figure 5).

### 2.5. PHB Complex as a Differentially Deregulated Mitochondrial Sensor in ALS and FTLD-U

In the proteomic phase, the down-regulation of Prohibitin-2 (*PHB2*) suggested a mitochondrial imbalance. This observation together with the mitochondrial imbalance revealed by the protein interactomes lead to a further evaluation and characterization of the *PHB* complex. Both *PHB1* and 2 were down-regulated across the two diseases in spinal cord, with a stronger down-regulation of *PHB2*. Interestingly, only for FTLD both *PHB1* and *PHB2* were found significantly dysregulated when analyzing NMC samples with a non-significant trend to down-regulation in ALS (Figure 6). These data indicated the ALS-FTLD spectrum impacts on the *PHB* complex leading to a possible mitochondrial dysfunction in spinal cord for ALS and FTLD-U and in NMC in the case of FTLD-U patients. These results would reinforce the hypothesis of mitochondrial dysfunction in ALS-FTLD spectrum hinted by the results obtained in the functional analysis.

## 3. Discussion

ALS and FTD share clinical features, anatomopathological characteristics, genetic mutations and pathway alterations leading to neurodegeneration. Consequently they are often presented as the two extremes of a common disease spectrum [15,16]. The present study aims to contribute to a better understanding of the overlapping and differential mechanisms underlying the development of the two different manifestations of these neurodegenerative processes. 

The experimental design of the present study was conceived to perform a deep proteomic analysis of the spinal cord from ALS-TDP-43 patients and FTLD-U patients (all of them in both groups without beta-amyloid, tau or alpha-synuclein inclusions) comparing them with post mortem tissue from the same region of non-neurodegenerative control donors, in order to deep in the tardopathies knowledge. The data resulting from the two comparative studies where afterwards confronted to evaluate similarities and differences among the two syndromes. 

In parallel a second objective was pursued; defining new diagnosis and prognosis potential biomarkers in ALS. To this aim the study included a technical validation in spinal cord and an additional cross-disease analysis in a non-motor cortex region carried out for a panel of selected proteins found significantly regulated in ALS. We hypothesize that protein expression changes detected in the anterior horn of the spine in ALS patients that can also be measured in FTLD-U patients, with no motor clinic may well be a reflection of preclinical neuropathogical alterations, indicating primary mechanisms involved in early stages of tardopathies. The idea of analyzing the expression of potential ALS biomarkers in non-motor cortex was conceived with the intention of exploring the regional involvement of these specific proteins, to determine whether they were exclusively regulated in motor regions or if also changes in non-motor cortex of the same proteins could derive in future development of FTLD-U.

The resulting quantitative proteomics data evidenced a more intense damage of the spine for ALS patients, with 281 proteins showing significant differential regulation in the anterior horn of the spine when comparing ALS and control post mortem tissue while only 52 differentially expressed proteins were for the FTLD-U patients (Figure 1). 33 out of these 52 proteins (more than 60%) where common differences to the ones described in the ALS, showing a common spectrum of regulation in this region of the CNS (Table 1). 

To perform an in silico validation of the proteomic and bioinformatics pipeline, a hypothesis driven approach was adopted to initially evaluate the robustness of the here obtained proteomic data. This approach confirmed that among the significantly dysregulated proteins for ALS, we could observe fourteen proteins that had previously been described as relevant in the ALS context. Among the significantly dysregulated proteins identified in this study, some of the best characterized biomarker candidates for ALS such as neurofilament heavy and medium polypeptides, cystatin C or *SOD1* (one of the best known ALS causing mutations) (Table 2) were found, thus corroborating the excellent quality of our data and supporting the reliability of the new observations here reported.

Among the candidates, proteins well characterized and widely described to be altered in ALS in previous studies, like Galectin-3 (*LGALS3*) Prealbumin (*TTR*) Protein S100-A6 (*S100A6*) and Protein S100-A11 (*A100A11*) were selected in order to reinforce the existing knowledge and literature related to them [32,34,46,47,48]. Four more proteins, Peptidyl-prolyl cis/trans isomerase (*PIN-1*), Selenium-binding protein 1 (*SELENBP1*), Calcyclin-binding protein (*CACYBP*) and Rho-associated protein kinase 2 (*ROCK 2*) (Figure 2), were selected as well to be retested, due to their potential novel role in ALS. 

Different regulation patterns were observed among the measured proteins *LGALS3* and *SELENBP1*, showing an area and disease specific regulation. Specifically, the up-regulation of both proteins was validated by western blot, partially confirming the quantitative LC-MS/MS approach used in this study. We could clearly validate the up-regulation observed in spinal cord for ALS patients for both proteins, with almost no changes in the expression of these two proteins in FTLD-U in that region. An important increase was also detected for both proteins in NMC, but exclusively for FTLD-U patients with no relevant changes for the ASL patients. These findings suggest that *LGALS3* and *SELENBP1* play a direct role in neuronal damage, being increased in the affected areas, for each specific disease but not in potentially pre-symptomatic areas. The up-regulation of these two proteins then could be disease specific and could participate in different neurodegenerative syndromes depending on the regional overexpression. *LGALS3* is a beta-galactosidase binding protein expressed by almost all cell type, involved in several physiological functions like immune activation and apoptosis [49,50,51]. *LGALS3* has been reported to show altered patterns of expression in different neurodegenerative diseases like Alzheimer disease (AD), Parkinson’s disease (PD) and ALS [34,52]. Elevated levels of *LGALS3* have been observed in previous proteomic approaches in CSF of ALS patients where this protein was proposed as a good potential biomarker for ALS [32,34]. Our study shows interesting regulation events for this protein, and there are previous data supporting this biomarker candidate also as a good therapeutic target. 

A proteomic study, complementary to the one presented here was recently published, the study presented a deep proteome mapping of the post-mortem frontal cortex and described protein differences along the ALS-FTD disease spectrum. Protein co-expression modules were built in the study, in order to represent changes in expression levels for modules associated with different processes. Among the differentially dysregulated modules they could see *SELENBP1* as hub protein for homeostatic processes, a module significantly regulated between control, ALS, ALS-FTD and FTD [27]. To our knowledge it is the only study that has reported direct evidence of *SELENBP1* involvement in ALS, nevertheless Glatt, S.J., et al. reported *SELENBP1* expression to be significantly up-regulated in post-mortem brain tissue from patients with schizophrenia and cognitive impairment [53]. In addition, the relationship between elevated selenium levels and ALS has been previously reported [54]. Finding region and disease specific *SELENBP1* elevated levels could suggest that *SELENBP1* is implicated in reducing levels of free selenium that are available for incorporation into selenoproteins, that have a cytoprotective effect and play a role in neuroprotection [55]. 

Disease specific regulation was observed for *TTR, S100A6, S100A11* and *PIN1*. These proteins were found significantly dysregulated in ALS and not in FTLD-U. *TTR* is a tetramer involved in blood transport of retinol and thyroxine. It has been suggested as a potential CSF biomarker in ALS and despite the role of *TTR* in the CNS remains understudied, it seems to be relevant in nerve regeneration and neurite/axonal outgrowth [46]. According to previous studies, motor neurons synthetize and secrete *TTR*, playing a role as a neuroprotective factor in AD and stroke [30,56,57]. The elevated *TTR* found in this study could be related to an activation of regenerative mechanisms as a response against the damage caused by ALS, supporting the role of *TTR* as potential biomarker in ALS.

*S100A6* and *SA100A11* were only detectable in spinal cord and both were significantly up-regulated in ALS but not in FTLD-U. S100 proteins have been related to Amyloid fibril formation, *S100-A6* up-regulation has been related to an increase in *SOD-1* aggregation [57]. This protein is also up-regulated in a *SOD-1* mouse model, showing overexpression in astrocytes in the anterior horn of the spine. *S100A6* seems to be specific for ALS, a valuable characteristic for a potential diagnostic marker [58].

*PIN-1* is one of the 33 proteins that was quantified as significantly down-regulated for both diseases in spinal cord in the proteomic analysis (Table 1). In the validation process, a significant down-regulation for ALS patients and a trend for down-regulation in FTLD-U patients was observed. Similar results were observed at non motor cortical level. *PIN-1* is highly expressed in neurons and it is involved in phosphorylation of neurofilaments (NFs). Aberrant phosphorylation of NFs, leads to accumulation and has been found to be very relevant in different neurodegenerative diseases, especially in ALS [59]. *PIN-1* has been described to be depleted in AD leading to an accumulation of phosphorylated tau protein [60]. Considering these discoveries together, *PIN-1* appears to be a plausible candidate to be a good marker of neurodegeneration with a particularly relevant involvement in protein aggregation and accumulation. Nevertheless to further develop this idea a wider study considering larger cohorts of patients and analyzing different regions of the CNS for different neurodegenerative diseases would be required. 

*CACYBP* and *ROCK2*, showed a regulation profile that appear to be neither disease nor area specific. *CACYBP* has never been directly linked to ALS, nonetheless there are many indirect links with other neurodegenerative diseases like Huntington disease, PD and AD [61,62]. *CACYBP* is known to be involved in cytoskeletal dynamics and in the regulation of transcptional response in neurons. Here in the proteomic analysis *CACYBP* was found to be decreased in the spinal cord for ALS patients. The western blot analysis confirmed the proteomic results for ALS patients in spinal cord and interestingly showed the opposite trend when analyzing NMC, revealing a significant up-regulation of this protein in both diseases in that specific region. A similar scenario was found when exploring *ROCK2* expression changes, no region or disease specificity was found (down-regulated for both syndromes in spinal cord and only for ALS in NMC). There are little evidence of Rho kinases being linked to ASL; however Rho kinase inhibition has been described to have a neuroprotective effect in a *SOD1* (G93A) mouse model of ALS [63] and abnormal expression of *ROCK2* has been associated with high levels of myosin binding protein H expression in ALS [64], making these two candidates new possible candidates for future study and additional exploration. Here we contribute with further evidence of the down-regulation of *ROCK2* in ALS and FTLD-U especially in spinal cord and also in NMC for ALS patients.

Network-driven proteomics is a straightforward approach to detect unexpected connections, considering that and with the aim to identify candidate ALS causative targets. We explored the *PHB* complex as a driver of the mitochondrial imbalance detected by the mass spectrometry analysis, both in ALS (Figure 3) and FTLD-U (Figure 4). An increasing number of studies are reporting clear functional evidence of impairment of the respiratory chain in ALS and other neurodegenerative disorders [65]. To explore this impairment in depth, we selected two mitochondrial proteins Prohibitin 1 and 2 (*PHB1* and *PHB2*) (Figure 6). *PHB*s have an important role in the assembly of subunits of mitochondrial respiratory chain complexes. The two *PHB* proteins, are located in the mitochondrial inner membrane where they form a large complex. *PHB2* was detected among the significantly down-regulated proteins in the proteomic study for the ALS patients, thus we found interesting to study in more detail the down-regulation of the *PHB* complex, a crucial mitophagy receptor and one of *TDP-43* [66] interacting partners in the mitochondria. Additionally, *PHB* complex is differentially modulated across several types of dementia [67]. In the present study *PHB1* and *PHB2* were analyzed in the two regions of interest, proving again the significant down-regulation of *PHB1* and very significant for *PHB2* in spinal cord for both diseases. Interestingly, both proteins were only found dysregulated for FTLD-U in NMC. This finding could suggest that mitochondrial damage happens early in this neurodegenerative events, and early stages not presenting motor symptoms are already suffering molecular alterations at mitochondrial level in the spinal cord as hinted by the functional analysis performed on the differential protein expression for both diseases.

In accordance with previous transcriptomic analysis performed in spinal cord for ALS subjects protein interactomes highlighted the involvement in cell death and survival signal regulation. There is evidence for abnormal regulation of protein kinases in several neurodegenerative diseases including ALS where altered activities and altered levels of specific kinases leads to abnormal phosphorylation and aberrant events that could be contributing to the pathogenic events [18,68]. 

The upstream signaling interactome of differentially expressed proteomes in the spinal cord and NMC of ALS and FTLD-U patients, revealed novel insights about the kinase dynamics present in the neurodegenerative motor neurons in ALS (Figure 3 and Figure 4). Significant down-regulation for *AKT/pAKT* and *FAK/pFAK* in the spinal cord both for ALS and FTLD-U patients was observed. Down-regulation for these signaling mediators, except for *pAKT* was also measured in NMC for FTLD-U while a non-significant up-regulation was assessed for ALS samples in this region. These data suggest a disease dependent regional depletion of these factors. *PI3-K/Akt* pathway is involved in the protective effect mediated by vascular endothelial growth factor (*VEGF*) that reduces mutant *SOD1*-mediated motoneuron death and enhances motoneuron survival [69,70,71], suggesting that the *PI3-K* signaling pathway plays a pivotal role in the survival of motoneuron [72]. Glutamate receptors act through many intracellular signaling pathways [73], and have been proposed as good potential treatment targets for ALS due to the key role played by Glutamate in neural development, and synaptic plasticity. *PI3-K* signaling pathway can be involved in the regulation of the mechanism of glutamate-induced excitotoxicity. This mechanism is not completely elucidated, nevertheless there is evidence of the mediation through the activation of mitogen-activated protein kinases (*MAPK*s) and inhibition of the *PI3K/AKT* pathways [74]. 

Respect to *MAPK* pathway *ERK/pERK* and MEK/*pMEK* systems were also evaluated. Significant regulation was observed for *MEK* in ALS and *pERK* and *pMEK* for FTLD-U. These results suggest that *MEK/ERK* signaling axis activation are more active in NMC during degeneration while some degree of depletion can be measured in spinal cord.

In conclusion, our study corroborates the overlap between ALS and FTD, sharing modifications in protein expression even in pre-symptomatic areas. These data could reflect the existence of several primary pathogenic mechanisms responsible for the initiation of neuronal damage in *TDP-43* proteinopathies. In addition to that we have confirmed the involvement of specific proteins previously associated with ALS as *LGALS3* and *TTR, S100A6, S100A11*, and have observed the involvement of other new proteins involved in ALS not previously described as *SELENBP1, PIN-1, CACYBP* and *ROCK2*. Additional targeted experiments are needed to functionally evaluate the role of this protein panel in tardopathies. 

Moreover, mitochondrial impairment and cell signaling pathway regulation has also been described and explored in detail to provide new insights of the involvement and relevance of PHB complex in the mitochondrial impairment and the interest of *AKT/pAKT* mediated signaling events in cell death and survival in ALS and FTLD-U. These results should be analyzed in larger samples.

## 4. Materials and Methods 

### 4.1. Patient Selection

Post-mortem fresh-frozen cervical spinal cord and frontal non-motor cortex tissue samples of 9 ALS patients, 8 FTLD-U and 8 age and gender matched controls, were obtained from the Biobank of Navarra (Navarrabiomed-FMS) following the guidelines of Spanish legislation [75] on the research matter with the approval of the Navarra Ethics Research Committee (2015/5, 2 February 2015). Procedures were in accordance with the Helsinki Declaration of 1975 as revised in 2000.

Patient selection was performed by expert neurologists and different inclusion criteria were used for each group involved in the study: For ALS patients, clinical and neuropathological diagnosis [76] of ALS with *TDP-43* deposition, absence of neuronal loss in frontal cortex and no clinical of dementia were required [77]. For the FTLD-U group: neuropathological diagnosis of FTLD-U [76] with positive *TDP-43* inclusions, absence of neuronal loss in spinal cord, no corticospinal tract degeneration and no clinical motor symptoms presenting individuals were included. Finally, for the control group, age and gender matched donors without neurodegenerative disease, cancer, recent vascular cerebral disease, infection or head injury neither family history of ALS or dementia were selected (demographic and clinical features reported in Table 3). 

### 4.2. Pathology and Immunohistochemistry

After brain autopsy, macroscopy examination and dissection trough middle line to separate both hemispheres were carefully done. Brain stem was obtained including medulla oblongata and spinal cord in all cases (Figure 7). Left hemisphere was placed in 10% formalin during 4 weeks and representative brain areas were selected as previously described [62]. Formalin-fixed, paraffin-embedded tissue sections from each region of interest were sectioned at 5 μm and counterstained with haematoxylin-eosin for immunohistochemistry analysis with the anti-phospho TDP-43 monoclonal antibody (1:80,000, p5409/410, Cosmo Bio, Otaru, Hokkaidō, Japan), mouse monoclonal antibody anti-human PHF-*TAU* (clone AT-8, Innogenetics, Ghent, Belgium), mouse monoclonal (S6F/3D) anti Beta-amyloid (Leica, Wetzlar, Germany) and mouse monoclonal antibody against α-synuclein (NCL-L-*ASYN*; Leica Biosystems, Wetzlar, Germany) and were visualized using an automated slide immunostainer (Leica Bond Max, Leica Bond Max) with Bond Polymer Refine Detection (Leica Biosystems Newcastle Ltd., Newcastle, UK). Luxol fast blue staining and CD 68 were included in brain stem sections for the study of myelin pathology. All ALS cases demonstrated upper and lower motor neuron degeneration accompanied by p-TDP43 neuronal inclusions. FTLD-TDP cases showed deposits in anterior cingulate cortex, limbic regions and absence in spinal cord area. They were classified onto one of four pathological subtypes (FTDL-TDP type A-D) using the recently updated classification system for FTLD-TDP pathology.

### 4.3. Sample Preparation for Proteomic Analysis

Anterior horns of the spinal cord samples were processed for protein extraction. Frozen Neurological post-mortem tissue samples were collected and homogenized in lysis buffer containing 7 M urea, 2 M thiourea and 50 mM DTT by mechanical disruption assisted by a Potter (Sartorius, Potter S, Goettingen, Germany). The resulting homogenates were ultracentrifuged at 100,000× *g* for 1 h at 15 °C. Prior to proteomic analysis, protein extracts were precipitated with methanol/chloroform, and pellets dissolved in 6 M Urea, Tris 100 mM pH 7.8. Protein quantitation was performed with the Bradford assay kit (Bio-Rad, Hercules, CA, USA) and 100 µg of each protein extract were subjected to enzymatic digestion using trypsin (Promega; ratio 1:50, *w*/*w*) at 37 °C for 16 h. Purification and concentration of peptides was performed using C18 Zip Tip Solid Phase Extraction (Millipore, Burlington, MA, USA).

### 4.4. Mass Spectrometry

Peptides mixtures were separated by reverse phase chromatography using an EksigentnanoLC ultra 2D pump fitted with a 75 μm ID column (Eksigent 0.075 × 250 mm). Samples were first loaded for desalting and concentration into a 0.5 cm length 100 μm ID precolumn packed with the same chemistry as the separating column. Mobile phases were 100% water 0.1% formic acid (FA) (buffer A) and 100% Acetonitrile 0.1% FA (buffer B). Column gradient was developed in a 240 min two step gradient from 5% B to 25% B in 210 min and 25% B to 40% B in 30 min. Column was equilibrated in 95% B for 9 min and 5% B for 14 min. During all process, pre-column was in line with column and flow maintained all along the gradient at 300 nL/min. Eluting peptides from the column were analyzed using an Sciex 5600 Triple-TOF system. Information data acquisition was acquired upon a survey scan performed in a mass range from 350 *m*/*z* up to 1250 *m*/*z* in a scan time of 250 ms. Top 35 peaks were selected for fragmentation. Minimum accumulation time for MS/MS was set to 100 ms giving a total cycle time of 3.8 s. Product ions were scanned in a mass range from 230 *m*/*z* up to 1500 m/z and excluded for further fragmentation during 15 s.

### 4.5. Data Analysis

Mass spectrometry raw data acquisition was performed using Analyst 1.7.1 (Sciex) and spectra files were searched employing Protein Pilot Software (v.5.0-Sciex), using Paragon™ algorithm (v.4.0.0.0) for database search, Progroup™ for data grouping, and searched against the concatenate target-decoy UniProt proteome reference Human database (Proteome ID: UP000005640, 70902 proteins, December 2015). False discovery rate was performed using a non-lineal fitting method and displayed results were those reporting a 1% Global false discovery rate or better at three different levels: spectral matching, peptide identification and protein inference

The peptide quantification was performed using Progenesis LC−MS software (ver. 2.0.5556.29015, Nonlinear Dynamics). Using the accurate mass measurements from full survey scans in the TOF detector and the observed retention times, runs were aligned to compensate for between-run variations in our nanoLC separation system. To this end, all runs were aligned to a reference, automatically chosen by the software, and a master list of features considering *m*/*z* values and retention times was generated. The quality of these alignments was manually supervised with the help of quality scores provided by the software. The peptide identification files were exported from Protein Pilot software and imported into Progenesis LC−MS software where they were matched to the respective features. Output data files were managed using R scripts for subsequent statistical analyses and representation. Proteins identified by site (identification based only on a modification), reverse proteins (identified by decoy database) and potential contaminants were filtered out. Proteins quantified with at least two unique peptides, a *t*-test *p*-value lower than 0.05, and an absolute fold change of <0.77 (down-regulation) or >1.3 (up-regulation) in linear scale were considered to be significantly differentially expressed.

### 4.6. Bioinformatics

All the resulting significant differences found on the proteomic study were further analyzed using QIAGEN’s Ingenuity^®^ Pathway Analysis (IPA) (QIAGEN Redwood City, www.qiagen.com/ingenuity) software, to identify and study differentially activated/deactivated pathways. This software comprises curated information from databases of experimental and predictive origin, enabling discovery of highly represented functions, pathways, and interaction networks. The IPA comparison analysis considers the signaling pathway rank according to the calculated *p*-value and reports it hierarchically. The software generates significance values (*p*-values) between each biological or molecular event and the imported molecules based on the Fisher’s exact test (*p* ≤ 0.05). 

### 4.7. Western-Blotting

Equal amounts of protein (10 μg) were resolved in 4–15% or 10–20% Criterion™ TGX Stain-Free™ Protein Gels (Bio-Rad, Hercules, CA, USA) (depending on the molecular weight of the target). Electrophoresis separated proteins were transferred into nitrocellulose membranes using Trans-Blot Turbo (Bio-Rad, Hercules, CA, USA) for 7 min at 2.5 A constant, up to 25 V. Equal loading of the gels was assessed by stain free digitalization for the experiments to validate changes detected in the proteomic study. Membranes were probed with primary antibodies at 1:1000 or 1:100 dilution in 5% non-fat milk or bovine serum albumin (BSA) (Appendix A). After incubation with the appropriate horseradish peroxidase-conjugated secondary antibody, antibody binding was detected by a Chemidoc™ MP Imaging System (Bio-Rad, Hercules, CA, USA) after incubation with an enhanced chemiluminescence substrate (Perkin Elmer, Waltham, MA, USA). All Band intensities were measured with Image Lab Software Version 5.2 (Bio-Rad, Hercules, CA, USA). Optical density values were expressed as arbitrary units and were normalized to total stain in each lane.

## Figures and Tables

**Figure 1 ijms-20-00004-f001:**
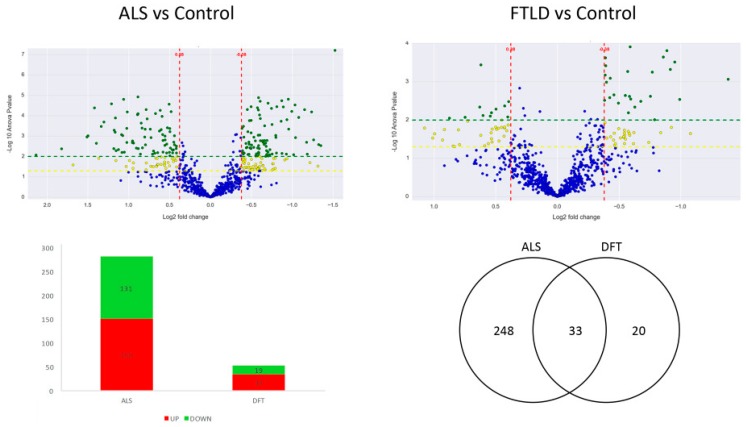
The two volcano plots are the graphical representation of the quantitative comparison performed in the present study. Each dot represents a protein; in blue unchanged proteins and in yellow (−log10 *p* value > 1.3) and green (−log10 *p* value > 2) the ones significantly dysregulated in each analysis. The first volcano plot shows the ALS vs control comparison and the second one shows the FTLD-U vs. control comparison. The Bar plot describes the number of significantly dysregulated proteins (up-regulated: red. Down-regulated: green). The Venn diagram illustrates the number of significantly dysregulated proteins in each disease and the observed overlap across comparisons.

**Figure 2 ijms-20-00004-f002:**
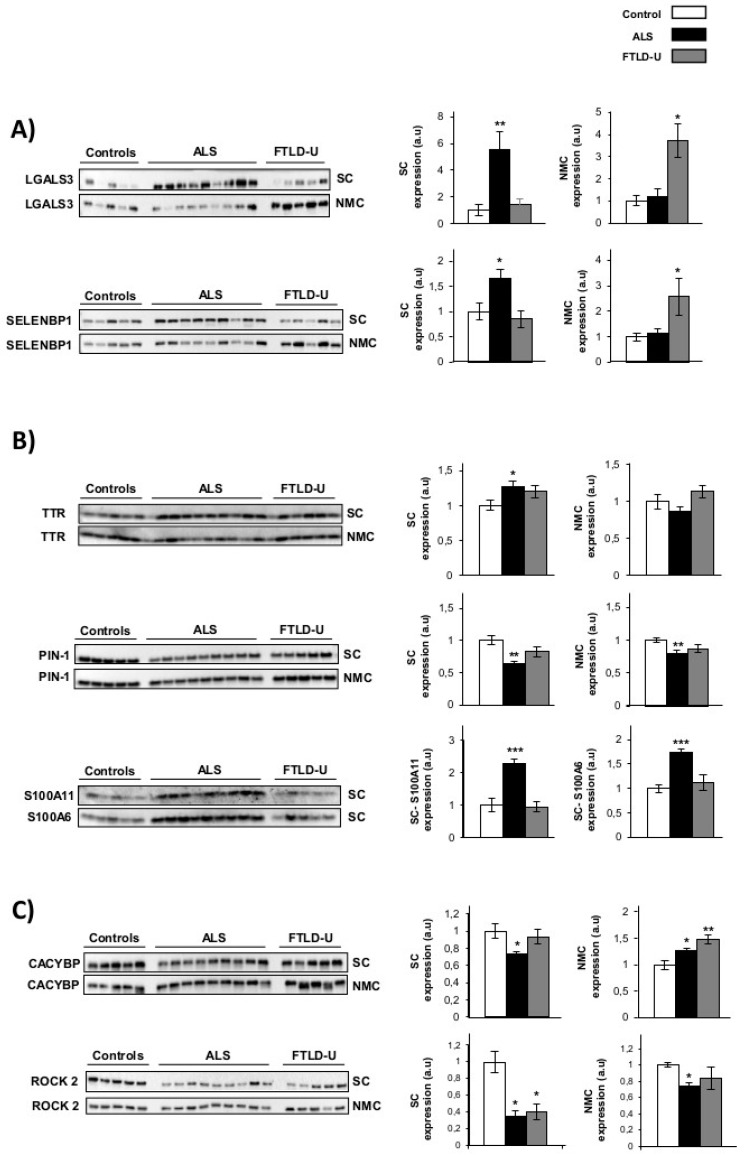
Western blot validations for dysregulated proteins of interest. Western blot analysis for the verification of expression changes for eight proteins identified as significantly dysregulated in the discovery proteomic study. (**A**) Area and disease specific regulation: *LGALS3* and *SELENBP1* (**B**) ALS specific regulation: *TTR, S100A11, S100A6* and *PIN1* (**C**) Not disease or area specific protein regulation; *CACYBP* and *ROCK2*. In each plot the optical density for control samples (white), ALS samples (black) and DFT samples (grey) are represented. Differential expression was evaluated in Spinal cord (SC) and Non motor cortex (NMC) for all the proteins under study except for *AS100 A11* and *AS100A6* that could only be measured in spinal cord. * *p* value < 0.05, ** *p* value < 0.01, *** *p* value < 0.001. a.u. arbitrary units.

**Figure 3 ijms-20-00004-f003:**
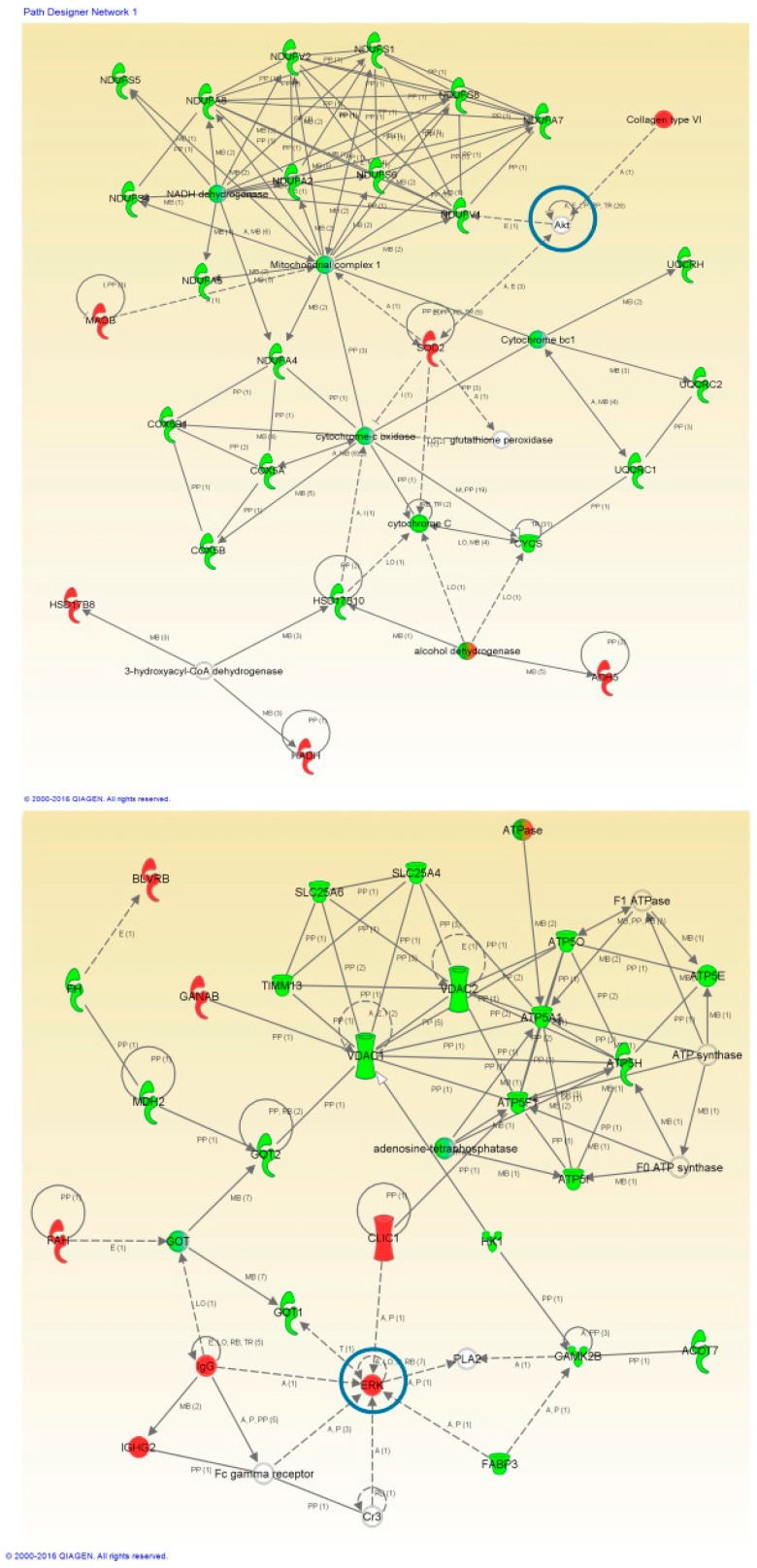
High-scoring protein interactome maps for differentially expressed proteins in the ALS versus control comparison. Dysregulated proteins are highlighted in red (up-regulated) and green (down-regulated). Continuous and discontinuous lines represent direct and indirect interactions respectively. The complete legend including main features, molecule shapes, and relationships can be found in http://ingenuity.force.com/ipa/articles/Feature_Description/Legend. In these visual representations of the relationships between differential expressed proteins we observe a significantly regulated protein network representing Mitochondrial and Metabolic Impairment in the first network of regulated proteins and Nucleic Acid Metabolism and Energy production related protein interaction in the second one. Proteins surrounded by a blue circle are involved in cell signaling.

**Figure 4 ijms-20-00004-f004:**
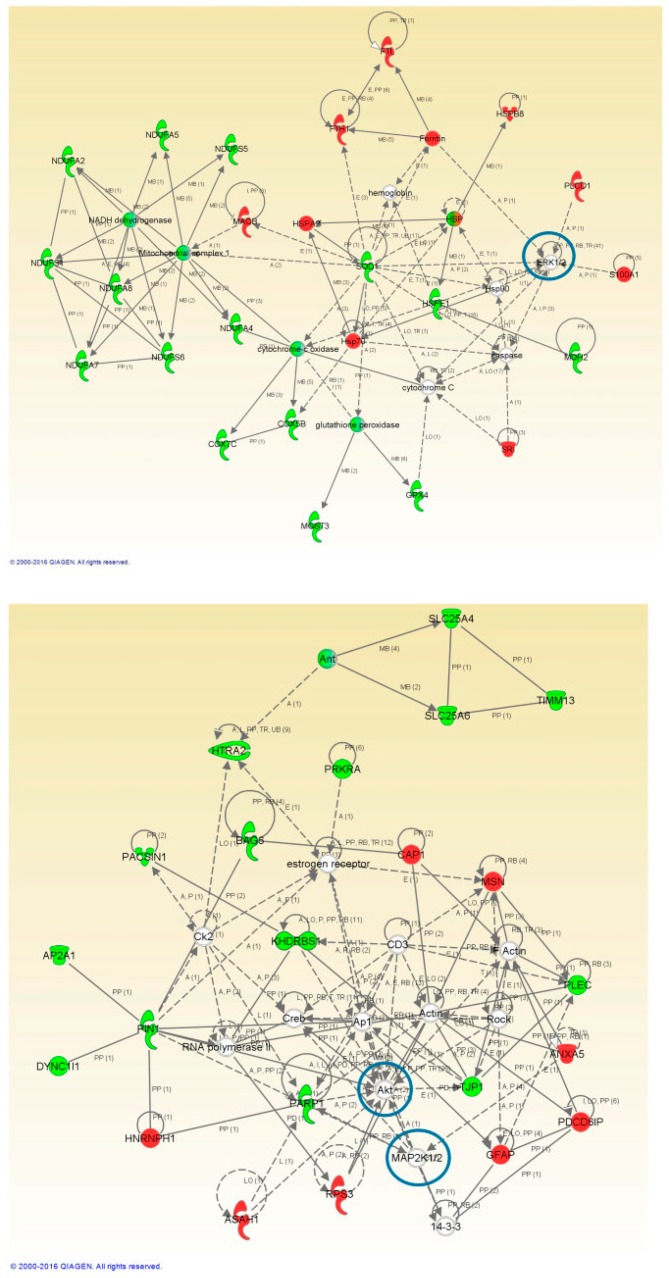
High-scoring protein interactome maps for differentially expressed proteins in the FTLD-U versus Control comparison. Dysregulated proteins are highlighted in red (up-regulated) and green (down-regulated). Continuous and discontinuous lines represent direct and indirect interactions respectively. The complete legend including main features, molecule shapes, and relationships can be found in http://ingenuity.force.com/ipa/articles/Feature_Description/Legend. In these visual representations of the relationships between differentially expressed proteins, a significantly dysregulated protein network representing Mitochondrial and Metabolic impairment and cell death and survival was observed. Proteins surrounded by a blue circle are involved in cell signaling.

**Figure 5 ijms-20-00004-f005:**
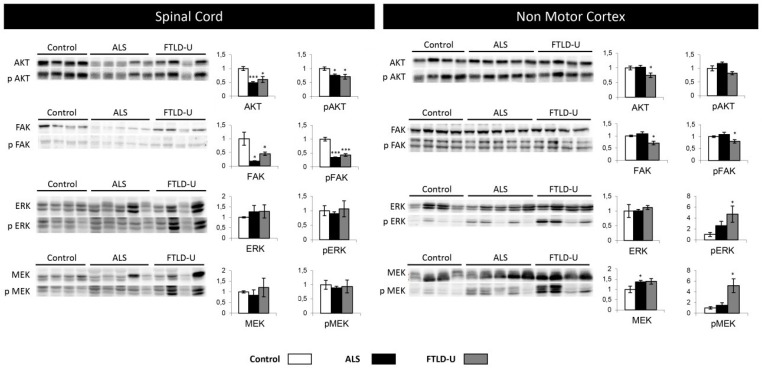
Cell signaling. Differential regulation in ALS and FTLD-U in different CNS regions. The left side of the figure shows the results for all the signaling proteins measured in spinal cord, while the right side of the figure shows the results obtained in non-motor cortex. In each plot (y axes) the optical density (in arbitrary units) is measured for control samples (white), ALS samples (black) and FTL-D samples (grey) are represented. * *p* value < 0.05, *** *p* value < 0.001.

**Figure 6 ijms-20-00004-f006:**
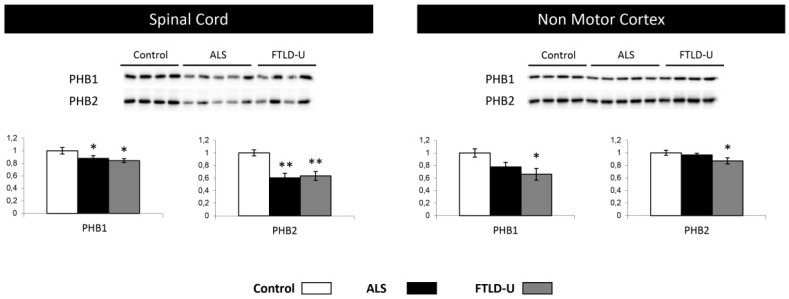
Mitochondrial impairment, *PHB1* and *PHB2* down-regulation. *PHB1* and *PHB2* were tested in spinal cord and Non motor cortex. Significant down-regulation for both proteins in both diseases was measured when analyzing spinal cord tissue, while significant down-regulation could be proved only in FTLD-U when analyzing Non motor cortex. In each plot (y axes) the optical density (in arbitrary units) is measured for control samples (white), ALS samples (black) and FTL-D samples (grey) are represented. * *p* value < 0.05, ** *p* value < 0.01.

**Figure 7 ijms-20-00004-f007:**
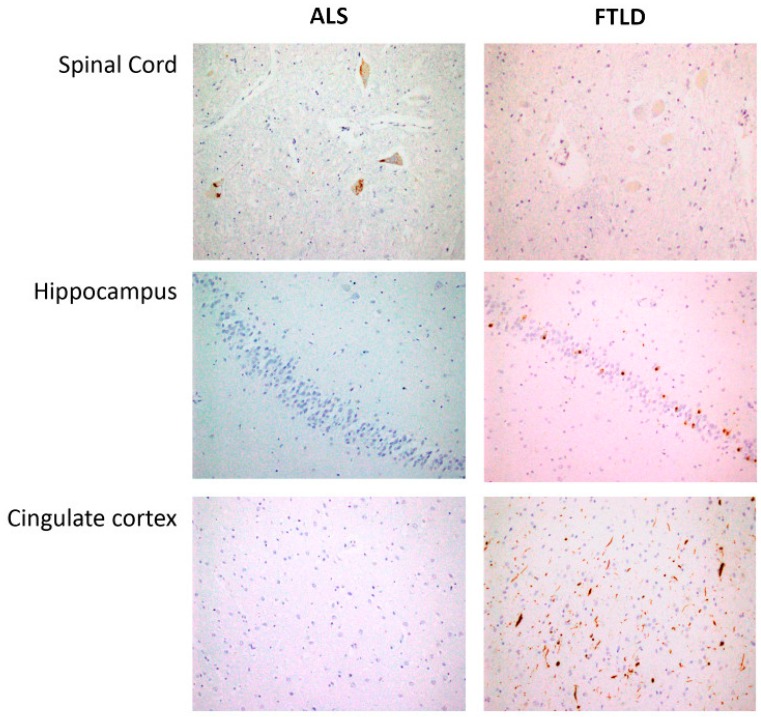
Spinal cord: Skein-like deposits of pTDP 43 in neurons of anterior horn from ALS patients and negative staining in neurons of anterior horn from FTLD patients (40×). Hippocampus: with negative staining in fascia dentata of hippocampus from ALS patients and intracytoplasmic inclusions of pTDP43 in hippocampus from FTLD patients (20×). Cingulate cortex: Negative staining in ALS patients and intracytoplasmic inclusions and long neurites of pTDP43 in FTLD patients (semantic dementia case) (20×).

**Table 1 ijms-20-00004-t001:** 31 out of the 33 proteins found significantly dysregulated both in ALS and FTLD-U are described here. Protein name, gene name, Uniprot code, number of unique peptides used for the identification and quantification as well as fold change and *p* value for the significantly dysregulated proteins in both diseases are shown in the table. The remaining two proteins were uncharacterized proteins (Uniprot code: C9JCJ5, K7N7A8) and are therefore not shown in this table.

Protein Name	Gene	Uniprot Code	Unique Peptides	*p*-Value ALS	*p*-Value FTLD-U	Fold-Change FTLD-U (log2)	Fold-Change ALS (log2)
**Common up-regulated proteins in spinal cord of ALS and FTLD-U patients**							
Protein kinase C and casein kinase substrate in neurons protein 1	*PACSIN1*	Q9BY11	7	0	0	−1.82	−0.59
Peptidyl-prolyl cis-trans isomerase NIMA-interacting 1	*PIN1*	Q13526	10	0	0	−0.99	−0.39
NADH dehydrogenase [ubiquinone] iron-sulfurprotein 6, mitochondrial	*NDUFS6*	O75380	7	0	0	−1.34	−0.56
NADH dehydrogenase [ubiquinone] 1 alpha subcomplex subunit 7	*NDUFA7*	O95182	4	0	0	−1.64	−0.95
Methylglutaconyl-CoA hydratase, mitochondrial	*AUH*	Q13825	5	0	0	−1.18	−0.77
Tubulin polymerization-promoting protein	*TPPP*	O94811	18	0	0.01	−1.17	−0.58
NADH dehydrogenase [ubiquinone] iron-sulfur protein 5	*NDUFS5*	O43920	3	0	0	−1.01	−0.88
ATP-dependent RNA helicase A	*DHX9*	Q08211	2	0	0.01	−0.73	−0.55
Isoform 2 of NADH dehydrogenase [ubiquinone] 1 alpha subcomplex subunit 5	*NDUFA5*	Q16718-2	4	0	0	−0.77	−0.43
Cytochrome b-c1 complex subunit 6, mitochondrial	*UQCRH*	P07919	5	0	0	−1.11	−0.55
MICOS complex subunit	*CHCHD6*	J3QTA6	4	0	0	−0.59	−0.59
MICOS complex subunit	*CHCHD3*	C9JRZ6	2	0	0.03	−0.98	−0.62
NADH dehydrogenase [ubiquinone] 1 alpha subcomplex subunit 2	*NDUFA2*	O43678	2	0	0	−1.45	−0.91
ATP synthase subunit d, mitochondrial	*ATP5H*	O75947	18	0	0	−0.55	−0.38
Cytochrome b-c1 complex subunit 7	*UQCRB*	P14927	4	0	0	−1.64	−1.59
ATP synthase subunit e, mitochondrial	*ATP5I*	P56385	4	0.01	0	−0.71	−0.5
d-tyrosyl-tRNA (Tyr) deacylase 1	*DTD1*	Q8TEA8	2	0	0	−0.62	−0.45
Mitochondrial import inner membrane translocase subunit Tim13	*TIMM13*	Q9Y5L4	4	0	0	−0.43	−0.39
Mitochondrial 2-oxoglutarate/malatecarrierprotein	*SLC25A11*	Q02978	2	0	0.01	−0.68	−0.56
ADP/ATP translocase 1	*SLC25A4*	P12235	2	0	0.01	−1.01	−0.61
Isoform 2 of Fructose-bisphosphate aldolase A	*ALDOA*	P04075-2	56	0.01	0	−0.62	−0.43
ARF GTPase-activating protein GIT1	*GIT1*	A0A0C4DGN6	2	0	0	−0.72	−0.88
**Common down-regulated proteins in spinal cord of ALS and FTLD-U patients**							
6-phosphogluconolactonase	*PGLS*	O95336	14	0	0	0.62	0.4
ATP-dependent 6-phosphofructokinase, muscle type	*PFKM*	P08237	8	0	0	0.6	0.63
Moesin	*MSN*	P26038	11	0	0.01	0.47	0.4
Guanine nucleotide-binding protein G(i) subunit alpha-2	*GNAI2*	P04899	1	0	0	1.15	1.05
Alcohol dehydrogenase class-3	*ADH5*	P11766	14	0	0.01	0.58	0.55
Annexin A5	*ANXA5*	P08758	9	0	0.01	1.12	0.61
Carbonic anhydrase 1	*CA1*	P00915	7	0	0.01	1.61	0.88
Small glutamine-rich tetratricopeptide repeat-containing protein alpha	*SGTA*	O43765	3	0	0	0.59	0.43
Heat shock protein beta-8	*HSPB8*	Q9UJY1	7	0	0.01	0.89	0.74

**Table 2 ijms-20-00004-t002:** Proteins found significantly dysregulated in the proteomic analysis and in the literature. All these 14 proteins have consistently been described as dysregulated in previous studies. Therefore our data reinforces the existing knowledge in ALS and the in silico validation shows the robustness of our study.

Gene	Uniprot	Protein Name	*p*-Value ALS	FC ALS	Molecular Function	Biological Function	ALS-Related
**Up-regulated proteins**							
*P4HB*	P07237	Protein disulfide-isomerase	0.00	1.07	ER foldase	ER Proteostasis	Mutations and enrichment [28]
*VCP*	P55072	Transitional endoplasmic reticulum ATPase	0.00	0.71	Multiple functions	DNA Repair/ER Proteostasis	Mutations and enrichment [29]
*S100A6*	P06703	Protein S100-A6	0.00	1.42	Ca^2+^/Zn^2+^ binding protein	calcium sensor and modulator	Enrichment [30]
*S100A11*	P31949	Protein S100-A11	0.00	2.08	Ca^2+^/Zn^2+^ binding protein	calcium sensor and modulator	Enrichment [31]
*LGALS3*	P17931	Galectin 3	0.01	0.52	Galactose-specific lectin	pre-mRNA splicing factor; acute inflammatory responses	enrichment (tissue, plasma and CSF) [32,33,34]
*TTR*	P02766	Prealbumin	0.00	1.37	Thyroid hormone-binding protein	thyroxine transport	Down-regulated in blood [35]
**Down-regulated proteins**							
*SOD1*	P00441	Superoxide dismutase [Cu-Zn]	0.05	−0.32	Multiple functions	Multiple functions	Mutations [36]
*INA*	Q16352	Alpha-internexin	0.01	−0.95	neuronal intermediate filament	Axonal structure and transport	Down-regulated in motor neurons [37]
*NEFM*	P07197	Neurofilament medium polypeptide	0.00	−0.71	neuronal intermediate filament	Axonal structure and transport	Down-regulated in CSF [38]
*NEFH*	P12036	Neurofilament heavy polypeptide	0.00	−0.97	neuronal intermediate filament	Axonal structure and transport	Up-regulated in CSF and Up in plasma [39,40]
*TUBA4A*	P68366	Tubulin alpha-4A chain	0.00	−1.20	Microtubules structure	Axonal transport	Mutations [41]
*CST3*	P01034	Cystatin-C	0.01	−0.67	cysteine protease inhibitor	Protein homeostasis	Down-regulated in CSF and up regulated in plasma [42,43]
*OPTN*	Q96CV9	Optineurin	0.00	−1.03	Multiple functions	Protein homeostasis and vesicle transport	Mutations and enrichment [44]
*VAPB*	O95292	Vesicle-associated membrane protein-associated	0.01	−0.60	Multiple functions	ER Proteostasis; vesicle transport; calcium homeostasis	Down-regulated in CSF [45]

**Table 3 ijms-20-00004-t003:** Clinical and demographic characteristics of all individuals under study. Relevant characteristics are described for the different patients and controls enrolled in the present study. Age, gender, age of diagnosis as well as presence of TDP-43 inclusions, motor neuron involvement, cognitive impairment and family background data for all the individuals under study were registered. Riluzole treatment and limbic or bulbar onset of the disease are also reported for ALS patients.

Pathological Diagnosis	Diagnostic Age	Exitus Age	Sex	TDP43	FTD	ALS	Motorneuron Involvement	Spinal Form	Bulbar Form	Cognitive Impairment	Family Background	Riluzole Treatment
Control	-	54	male	−	−	−	−	−	−	−	−	−
Control	-	26	male	−	−	−	−	−	−	−	−	−
Control	-	91	female	−	−	−	−	−	−	−	−	−
Control	-	103	male	−	−	−	−	−	−	−	−	−
Control	-	72	male	−	−	−	−	−	−	−	−	−
Control	-	91	male	−	−	−	−	−	−	−	−	−
Control	-	66	male	−	−	−	−	−	−	−	−	−
Control	-	88	female	−	−	−	−	−	−	−	−	−
ALS	56	59	male	+	−	+	+	+	−	−	−	+
ALS	71	73	female	+	−	+	+	+	−	−	−	+
ALS	54	61	female	+	−	+	+	+	−	−	−	+
ALS	66	69	female	+	−	+	+	−	+	−	−	+
ALS	67	69	male	+	−	+	+	−	+	−	−	+
ALS	47	49	male	+	−	+	+	+	−	−	−	+
ALS	71	79	male	+	−	+	+	+	−	−	−	+
ALS	61	63	male	+	−	+	+	+	−	−	−	+
ALS	25	40	female	+	−	+	+	+	−	−	−	+
FTLD-U	81	88	female	+	+	−	−	−	−	+	−	−
FTLD-U	68	77	male	+	+	−	−	−	−	+	−	−
FTLD-U	76	83	female	+	+	−	−	−	−	+	−	−
FTLD-U	58	73	male	+	+	−	−	−	−	+	−	−
FTLD-U	unknown	60	female	+	+	−	−	−	−	+	−	−
FTLD-U	79	87	male	+	+	−	−	−	−	+	−	−
FTLD-U	74	84	female	+	+	−	−	−	−	+	−	−
FTLD-U	77	85	male	+	+	−	−	−	−	+	−	−

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
