# Peer review of "Neuroanatomical Quantitative Proteomics Reveals Common Pathogenic Biological Routes between Amyotrophic Lateral Sclerosis (ALS) and Frontotemporal Dementia (FTD)"

_ijms, 2018, doi:10.3390/ijms20010004_

Reviewer 1 Report

Iridoy et al describe quantification of LCMS-MS quantification of ~1000 proteins in spinal cord from ALS (n=9), FTD (n=8) and control (n=8) subjects.  The information provided by the study is useful however there are a number of concerns.  The small sample numbers and the absence of functional validation studies limit the impact of the work.

Certain proteins are identified as differentially expressed which have been identified in previous studies which is a good validation of this work; other protein changes are novel.

A significant interesting aspect of this study is the examination of the overlap between changes in ALS and FTD and between spinal cord and frontal cortex.

Major points:

- The sample numbers are small leading to some concern about reproducibility.  The addition of testing in a validation cohort would be a significant improvement.

- Validation is performed by immunoblot using tissue from spinal cord and frontal cortex.  This does not conclusively demonstrate disruption of function.  However, in places the authors state that they have achieved this - for example: "the ALS‐FTLD spectrum impacts on the PHB complex leading to a mitochondrial dysfunction in spinal cord."  The language should be changed to make it clear that mitochondrial function has not been directly measured.  

- Perhaps one way of validating a functional effect of the changes observed would be to check which changes in protein expression correlate significantly with counts of TDP-43 pathology which is known to predict neuronal loss.  

- The language describing the statistical testing is not clear - the authors mention a FDR threshold of 1% but it appears that all proteins with a p<0.05 (t-test) were included.  Given the number of statistical tests performed (~1000), some multiple testing correction should be performed.  

- A way to improve power to detect disease-associated dysfunction is using a network level analysis.  The authors have performed this in the form of IPA.  However, the statistical significance of stated functional enrichments are not clearly stated - it would be useful to list enriched pathways and their significance level in a table.  

Minor:

- The language is difficult to understand in placed.  In particular, could 'regulated' be changed to 'dysregulated' when referring to disease-associated changes.

Author Response

In response to the Reviewer valuable comments, we have corrected and improved our manuscript. The changes will be highlighted  in the new version of the manuscript.

Major points:

1-     The sample numbers are small leading to some concern about reproducibility. The addition of testing in a validation cohort would be a significant improvement

The reviewer is right in pointing this out and  it would have been ideal to have a larger sample. However taking into account the relative low prevalence of these diseases and the strict neuropathological and clinical inclusion criteria we have used in our study, we consider that the number of samples is acceptable for a Discovery experiment in proteomics. We agree that the discoveries presented in this study must be validated in the near future in a larger sample size, which will be possible thanks to the growing generous and highly valuable patient donations.

2-                   Validation is performed by immunoblot using tissue from spinal cord and frontal cortex. This does not conclusively demonstrate disruption of function. However, in places the authors state that they have achieved this- for example: “the ALS-FTLD spectrum impacts on the PHB complex leading to a mitochondrial dysfunction in spinal cord.” The language should be changed to make it clear that mitochondria function has not been directly measured

Thank you for this helpful criticism. Our revised manuscript will contain some more appropriate language modifications.

3-                   Perhaps one way of validating a functional effect of the changes observed would be to check which changes in protein expression correlate significantly with counts of TDP-43 pathology which is known to predict neuronal loss.

We find that a very interesting observation and despite, at the moment, it remains unclear  how the TDP-43 aggregation leads to neuronal loss, the reviewer’s theory would be of great interest if it could be demonstrated a correlation between some protein expression with counts of TDP-43.

4-     The language describing the statistical testing is not clear-the authors mention a FDR threshold of 1% but it appears that all proteins with a p>0.05 (t-test) were included. Given the number of statistical tests performed (~1000), some multiple testing correction should be performed

FRD threshold of 1% is applied to identifications of peptides and inference of those peptides in protein groups, resulting in the final list of proteins identified. So, this FDR correction is not applied as multiple testing correction in the statistical t-test used. Pascovici el at (Pascovici et al. Proteomics 2016) demonstrated that multiple testing corrections are a useful tool for restricting the FDR, but can be blunt in the context of low power. Unfortunately, in proteomics experiments low power can be common, driven by proteomics-specific issues like small effects due to ratio compression, and few replicates due to reagent high cost, instrument time availability and other issues; in such situations, most multiple testing corrections methods, if used with conventional thresholds, will fail to detect any true positives even when many exist. In this low power, medium scale situation, other methods such as effect size considerations or peptide-level calculations may be a more effective option. In this study, we applied FDR 1% at level of spectral matching, peptide identification and protein inference.

In the text, the follow sentence has been added:

False discovery rate was performed using a non-lineal fitting method and displayed results were those reporting a 1% Global false discovery rate or better at three different levels: spectral matching, peptide identification and protein inference.

5-     A way to improve power to detect disease-associated dysfunction is using a network level analysis. The authors have performed this in form of IPA. However, the statistical significance of stated functional enrichments are not clearly stated-it would be useful to list enriched pathways and their significance level in a table

We completely agree with the reviewer and have now included a supplementary table listing all the pathways regulated in both syndromes. Supplementary table 1 now shows the significance level (-log p value > 1.3 equal to p-value < 0.05) and the significantly regulated genes measured for each specific Ingenuity canonical pathway found enriched in this study.

We have added the following in the revised manuscript:

Among the dysregulated features 21 Ingenuity canonical pathways were found significantly enriched, both in ALS and FTLD-U (Supplementary material table 1) among them Mitochondrial dysfunction was the most significantly enriched pathway for both diseases.

Supplementary table 2: Shows the 21 Ingenuity pathways that were found to be significantly regulated (-log p-value >1.3) when comparing ALS and FTLD-U with control subjects. The column named Molecules contains the gene names for the molecules found significantly regulated involved in each enriched pathway.

Minor points

1-      The language is difficult to understand in place. In particular, could “regulated” be changed to “dysregulated” when referring to disease-associated changes.

Thank you. We have made the suggested language corrections.

Reviewer 2 Report

This paper does some interesting analysis to compare ALS and FTLD, which  could be valuable to the field.  However, there are some changes and clarifications that need to be made prior publication.

While the experimental methods appears mostly sound and well-presented, there are some concerns with data analysis.  There is no mention as to whether data distributions were assessed for normality (Shapiro Wilks or another appropriate technique) prior to selection of statistical test.  Also, it states t-tests were used to compare means, but there was no mention of correction factors to the p-value for multiple comparisons, and the paper did have multiple comparisons.  Failure to correct the p-value by either using a correction factor (Bonferroni, etc.) OR selecting a more appropriate statistical test for multiple comparisons is needed to insure there are not false positives.

The figures are not high quality and appear very blurred.  Not sure if this is a function of the pdf printing or not.  If not, the authors need to make higher resolution figures for the final paper.  Figure 1, 3, and 4 is the worst in terms of clarity (and especially the network figures which are not even legible), but all figures should be examined for improvements of resolution.  Increasing font size a bit might help if resolution cannot be improved.

Further discussion on the neuropathological spectrum of disease both clinically and biochemically would help the introduction and/or discussion.  While the emphasis and key results are on ALS and FTLD, the results do loosely support even some overlap with Alzheimer’s and Parkinson’s.  In any case, a good neuropathology spectrum reference to add to his work and even potentially compare results to would be, Coan 2015, J Neurodegen Dis 2015;15(5):301-12. doi: 10.1159/000433581.

Author Response

In response to the Reviewer valuable comments, we have corrected and improved our manuscript. The changes will be highlighted  in the new version of the manuscript.

1-     While the experiment methods appears mostly sound and well-presented, thera are some concerns with data analysis. There is no mention as to whether data distributions were assessed for normality (Shapiro Wilks or another appropriate technique) prior to selection of statistical test. Also, it states t-tests were used to compare means, but there was no mention of correction factors to the p-value for multiple comparisons, and the paper did have multiple comparisons. Failure to correct the p-value by either using a correction factor (Bonferroni, etc.) OR selecting a more appropriate statistical test for multiple comparisons is needed to insure there are not false positives.

Pascovici el at (Pascovici et al. Proteomics 2016) demonstrated that multiple testing corrections are a useful tool for restricting the FDR, but can be blunt in the context of low power. Unfortunately, in proteomics experiments low power can be common, driven by proteomics-specific issues like small effects due to ratio compression, and few replicates due to reagent high cost, instrument time availability and other issues; in such situations, most multiple testing corrections methods, if used with conventional thresholds, will fail to detect any true positives even when many exist. In this low power, medium scale situation, other methods such as effect size considerations or peptide-level calculations may be a more effective option, even if they do not offer the same theoretical guarantee of a low FDR. In this case, we applied FDR 1% at level of spectral matching, peptide identification and protein inference. 

In the revised manuscript the follow sentence has been added:

False discovery rate was performed using a non-lineal fitting method and displayed results were those reporting a 1% Global false discovery rate or better at three different levels: spectral matching, peptide identification and protein inference.

2-                   The figures are not high quality and appear very blurred. Not sure if this is a function of the pdf printing or not. If not, the authors need to make higher resolution figures for the final paper. Figure 1,3 and 4 is the worst in terms of clarity (and especially the network figures which are not even legible), but all figures should be examined for improvements of resolution. Increasing font size a bit might help if resolution cannot be improved.

We thank the reviewer for the correction. Higher resolution figures have been made and we hope improvements of resolution will now be up to the journal standards.

3- Further discussion on the neuropathological spectrum of disease both clinically and biochemically would help the introduction and/or discussion. While the emphasis and key results are on ALS and FTLD, the results do loosely support even some overlap with Alzheimer’s and Parkinson’s. In any case, a good neuropathology spectrum reference to add to his work and even potentially compare results to would be, Coan 2015, J Neurodegen Dis 2015; 15(5):301-12

We thank the reviewer for the comments and, taking the suggestion into account, we have read with interest the recommended paper and we have added the indicated reference (ref. 12) in the revised manuscript. However the objective of our study was to deep in  the pathobiology of tardopathies and to emphasize this, we have added in the revised manuscript some clarifying comments regarding the neuropathology of our samples.

Reviewer 3 Report

This paper explores biological relationship between amyotrophic lateral sclerosis (ALS) and frontotemporal dementia (FTD). There have been many studies for exploring the relationship between these two diseases in the literature. This study recruits three groups of samples including ALS patients, FTD patients and control for proteomics study. There are 281 and 52 proteins found differentially‐expressed in ALS cases and in frontotemporal lobar degeneration cases when confronted to healthy control cases, respectively. 33 proteins were found to be significantly deregulated in both diseases.

“Frontotemportal” in the title should be “Frontotemporal”.

Could you release the raw data or the data is available when readers request it?

You said there are 33 proteins found to be significantly regulated in both diseases, but there are only 31 proteins in Table 1.

Line 83. “Proteomics”→ “proteomics”

Does the question mark “?” in Table 3 mean that the age of this patient is unknown? If yes, you can write “unknown” instead of “?”.

Author Response

In response to the Reviewer valuable comments, we have corrected and improved our manuscript. The changes will be highlighted  in the new version of the manuscript.

1-     “Frontotemportal” in the title should be “Frontotemporal”

Thank you. We have corrected it

2-    Could you release the raw data or the data is available when readers request it?

We agree with the reviewer on the importance of these data being available. We will add this data in the revised manuscript (Supplementary table 1) In this table protein name, uniprot code, gene name p-value and Fold change (FC) for each comparison are shown, for 299 of the 301 significantly dysregulated proteins found in both comparisons (p-value<0.01, fold="" change="">+/-0.37) ALS versus control and FTLD-U versus control. Two  uncharacterized proteins have been excluded from this list.

3-     You said there are 33 proteins found to be significantly regulated in both diseases, but there are only 31 proteins in Table 1.

The Reviewer is right. The remaining 2 proteins are uncharacterized proteins and we decided to leave them out because we considered only the 31 well-characterized were of interest for the reader. This will be explained in the figure legend in the revised manuscript.

Table 1. 31 out of the 33 proteins  found significantly regulated both in ALS and FTLD-U are described here.  protein name, gene name, Uniprot code, number of unique peptides used for the identification and quantification as well as fold change and p value for the significantly regulated proteins in both diseases are shown in the table. The remaining two proteins were uncharacterized proteins (Uniprot code: C9JCJ5, K7N7A8) and are therefore not shown in this table

4-     Line 83. “Proteomics” ”→ “proteomics”

Thank you. The change has been made

5- Does the question mark “?” in Table 3 mean that the age of this patient is unknown? If yes, you  can write “unknown” instead of “?”.

We agree with the reviewer and we have made the correction in the revised manuscript. This is surely a mistake or forgetfulness in some of the corrections that were made in the different versions.

Round  2

Reviewer 2 Report

I do think the statistical considerations are important for this work, and personally I am not fully convinced of the methods proposed despite literature citation and use by others, even in the proteomics field, as there are more advanced methods to examine multiple comparisons with better sensitivity and specificity within a network of relationships. Nonetheless, clarifications did help bring the methods into context for the reader to make their own decision, so I approve for publication. An examination of more advanced network alternatives or ML classification algorithms may be helpful for future related work.